# Ubiquitin B, Ubiquitin C, and β-Catenin as Promising Diagnostic and Prognostic Tools in Prostate Cancer

**DOI:** 10.3390/cancers16050902

**Published:** 2024-02-23

**Authors:** Daria Piątkowska, Anna Klimaszewska-Wiśniewska, Alicja Kosińska, Radosław Wujec, Dariusz Grzanka, Justyna Durślewicz

**Affiliations:** Department of Clinical Pathomorphology, Faculty of Medicine, Collegium Medicum in Bydgoszcz, Nicolaus Copernicus University in Torun, 85-094 Bydgoszcz, Poland; daria.piatkowska@cm.umk.pl (D.P.); ania.klimaszewska@op.pl (A.K.-W.); radoslaw.wujec@cm.umk.pl (R.W.); d_grzanka@cm.umk.pl (D.G.)

**Keywords:** prostate cancer, ubiquitination, UBB, UBC, β-Catenin, prognostic markers

## Abstract

**Simple Summary:**

Prostate cancer (PC) is a major global health concern, ranking as the second most common malignancy in men. This study investigates the role of ubiquitin B (UBB), ubiquitin C (UBC), and β-Catenin in PC development, focusing on their expression and interactions. By analyzing their profiles in PC tissues and lymph node metastases, the research identifies elevated UBB and UBC levels, particularly in metastatic cases, emphasizing their crucial roles in disease progression. Furthermore, the study reveals a correlation between high UBB and UBC expression and reduced overall survival in PC patients. Additionally, abnormal β-Catenin expression is associated with shorter survival and serves as a significant independent prognostic factor. The combined assessment of UBB, UBC, and β-Catenin emerges as a valuable approach for predicting patient outcomes in PC.

**Abstract:**

Prostate cancer (PC) is a major global public health concern, imposing a significant burden on men and ranking as the second most prevalent malignancy. This study delves into the intricate world of ubiquitination processes and expression regulation, with a specific focus on understanding the roles of ubiquitin B (UBB), ubiquitin C (UBC), and β-Catenin in PC development. We thoroughly analyze the expression profiles of UBB, UBC, and β-Catenin, investigating their interactions and associations with clinical and histopathological data. These findings offer valuable insights into their potential as robust prognostic markers and their significance for patient survival. Our research uncovers the upregulation of UBB and UBC expression in PC tissues, and an even more pronounced expression in lymph node metastases, highlighting their pivotal roles in PC progression. Moreover, we identify a compelling correlation between high UBB and UBC levels and diminished overall survival in PC patients, emphasizing their clinical relevance. Additionally, we observe a significant reduction in membranous β-Catenin expression in PC tissues. Importantly, abnormal β-Catenin expression is strongly associated with shorter survival in PC patients and serves as a significant, independent prognostic factor for patient outcomes. Kaplan–Meier survival analysis indicates that patients with tumors characterized by simultaneous UBB and aberrant β-Catenin expression exhibit the poorest overall survival. These collective insights underline the clinical importance of evaluating UBB, UBC, and β-Catenin as combined prognostic markers in PC.

## 1. Introduction

In 2020, prostate cancer (PC) was diagnosed in 1,414,259 men and was the cause of death in 375,304 cases, making it the second most common cancer type in the male population after lung cancer [1]. Elder men are more likely to be diagnosed with PC and this risk increases significantly after the age of 55 [2]. In the majority of cases, PC is detected at an early stage, which translates into a high probability of full recovery. Survival rates for patients diagnosed and treated in the initial stage of the carcinoma reach about 97.5%. This percentage decreases steeply to about 30% for more advanced cases, as well as when metastases are present [3,4]. PC progresses through stages, starting with prostate gland cell abnormalities and slow enlargement, followed by elevated PSA levels, non-castration metastasis, and potentially fatal castration-resistant conditions. Metastases, commonly starting in lymph nodes, can spread to organs, reducing quality of life and increasing mortality risk. Regrettably, metastatic PC remains generally incurable despite medical advances [4].

In recent years, scientific interest in understanding the role of ubiquitination processes and gene expression regulation in the context of PC development has grown significantly [5,6,7,8]. These studies have shed light on the potential roles played by ubiquitin B (UBB) and ubiquitin C (UBC) proteins, as well as β-Catenin, within the complex landscape of tumorigenesis associated with PC.

UBB’s involvement in the ubiquitination of β-Catenin underscores a critical regulatory mechanism within the cell. Ubiquitination is a post-translational modification that plays a pivotal role in maintaining protein homeostasis and controlling various cellular processes. In the case of β-Catenin, a key player in cell adhesion and the Wnt signaling pathway, ubiquitination serves as a finely tuned mechanism for its degradation and regulation. When the cell needs to reduce the levels of β-Catenin, such as to prevent uncontrolled cell growth or aberrant signaling, it orchestrates a process where β-Catenin is “tagged” with ubiquitin molecules. This tagging essentially earmarks β-Catenin for destruction, signaling the cellular machinery to target it for proteasomal degradation [9,10]. The proteasome acts as the cell’s waste disposal system, breaking down ubiquitinated proteins into their constituent parts [11]. UBC, functioning as a ubiquitin-conjugating enzyme, plays a central role in this process [12]. It facilitates the transfer of ubiquitin molecules to β-Catenin, thereby enabling the tagging process to occur. In essence, UBC acts as a molecular courier, shuttling ubiquitin molecules to their destination on β-Catenin [13]. Moreover, UBC can also collaborate with other E3 ligases, which are enzymes responsible for the specific recognition of target proteins, to ensure the efficient ubiquitination of β-Catenin [14,15]. This intricate regulatory network highlights the cell’s ability to finely tune the levels of β-Catenin, which is vital for maintaining normal cellular functions. Dysregulation in this process, as observed in various cancers, can lead to the accumulation of β-Catenin and abnormal activation of the Wnt signaling pathway, ultimately contributing to tumorigenesis and cancer progression [16,17]. Therefore, understanding the roles of UBB and UBC in β-Catenin ubiquitination is of paramount importance in deciphering the molecular mechanisms underlying cancer development and exploring potential therapeutic targets [18].

In our study, we investigate the levels of UBB, UBC, and β-Catenin, as well as their mutual correlations. Additionally, we examine the association between these factors and clinical and histopathological data to assess their prognostic significance by correlating expression levels with patient survival indicators.

## 2. Materials and Methods

### 2.1. Patients and Tissue Specimens

Archival formalin-fixed paraffin-embedded (FFPE) tissue samples were obtained from the patients with a diagnosis of PC. The cohort included 67 patients, who underwent a prostatectomy at the General and Oncological Urology Clinic, Antoni Jurasz University Hospital No. 1 in Bydgoszcz (Poland) between 2016 and 2019. Immunohistochemical staining was performed on tissue microarrays (TMAs) prepared from regions representative of the tumor (containing at least 80% tumor cells). Each recipient block contained fifteen distinct tissue fragments sourced from donor paraffin blocks (three cores from one patient). Additionally, blocks were constructed containing healthy tissue adjacent to the tumor from the same patients, as well as blocks with tissue from lymph node metastases. The analysis of lymph node metastases (*n* = 18) is specifically conducted on samples derived from the same patients as those with primary tumors. The analysis only considered the lymph nodes excised during prostate resection. The detailed clinical characteristics of patients are listed in Appendix A.

Survival time was defined as the time elapsed from the date of diagnosis to the date of death or last follow-up in case of cancer survivor patients. The follow-up time from prostatectomy was 9 to 2576 days with a median of 1787 days. The study is based on the approval of The Ethics Committee of Nicolaus Copernicus University in Toruń, Ludwik Rydygier Collegium Medicum in Bydgoszcz (approval number 248/2019).

### 2.2. Immunohistochemical Staining

The TMA blocks were cut using a manual rotary microtome (Accu-Cut, Sakura Finetek, Torrance, CA, USA) into 4.0 μm thick sections, which were placed on extra adhesive slides (Superfrost Plus; Menzel-Glaser, Braunschweig, Germany) and then dried. An immunohistochemical staining was performed using the BenchMark^®^ ULTRA automated slide processing system (Ventana Medical Systems, Tucson, AZ, USA). As a first step, deparaffinization and rehydration were conducted in EZ Prep solution (Ventana Medical Systems) at 72 °C for 8 min. Subsequently, an antigen retrieval was carried out in a high pH Cell Conditioning solution (CC1) for 64 min. An incubation was performed with the following antibodies, each at their respective dilutions: anti-UBB rabbit polyclonal antibody (cat. no: HPA049132, Sigma-Aldrich, St. Louis, MO, USA) at 1:100, anti-UBC rabbit polyclonal antibody (cat. no: HPA041344, Sigma-Aldrich, St. Louis, MO, USA) at 1:50, and anti-β-Catenin mouse monoclonal antibody (cat. no: 760-4242, Ventana Medical Systems, Tucson, AZ, USA) at a ready-to-use concentration, for a total incubation period of 32 min. In the next step, an ultraView DAB Detection Kit (Ventana Medical Systems) was used for visualization of the reaction. After that, the slides were counterstained in Bluing Reagent and Hematoxylin. Finally, the sections were dehydrated in increasing ethanol concentrations (80%, 90%, 96%, and 99.8%), cleared in xylenes, then mounted using mounting medium (Shandon Consul-Mount, Thermo Scientific, Waltham, MA, USA), and assessed under the microscope.

Additionally, concurrent labeling of UBB and β-Catenin was performed on the tumor section and normal tissue. The UBB antigen–antibody complex was visualized as a brown reaction product using the ultraView DAB Detection Kit (Ventana Medical Systems). To localize β-Catenin antigens in the same tissue, the ultraView Universal Alkaline Phosphatase Red Detection Kit (Ventana Medical System) was employed, resulting in a red reaction product for the β-Catenin antigen-antibody complex.

### 2.3. Evaluation of Immunohistochemistry Staining

Our slides were digitized using a whole slide imaging scanner (Roche Ventana DP 200) and were evaluated by both an image scientist and a pathologist. This evaluation was conducted using the modified Remmele–Stegner Index (IRS) scale. The final score, ranging from 0 to 12, is derived from the multiplication of two factors: the percentage of cells/areas stained positively (ranging from 0 to 4) and the staining intensity (ranging from 0 to 3). The scans were subsequently saved in our internal repository. The Evaluate Cutpoints application was utilized to determine the cutpoints. The threshold values for high and low membranous expression of β-Catenin were as follows: <8 (abnormal); ≥8 (normal). The nuclear expression of UBB and UBC was analyzed based on the presence or absence of nuclear staining in cells. The threshold values for membranous expression of β-Catenin were as follows: <8 (abnormal); ≥8 (normal). In turn, a cutpoint of ≥1 was used to indicate the presence of nuclear expression of UBB and UBC (<1, absent). In dual-IHC staining, nuclear UBB expression and membrane expression of β-Catenin were assessed based on the presence and absence of staining to test the hypothesis of the present study.

### 2.4. In Silico Analysis

#### 2.4.1. The Cancer Genome Atlas

To complement our findings and compare them with publicly available data, we additionally assessed expression of UBB, UBC, and CTNNB1 in The Cancer Genome Atlas (TCGA) cohort. Gene expression data for a cohort consisting of 495 patients with prostate adenocarcinoma and 100 non-tumor prostate tissues were obtained from the UCSC Xena Viewer (http://xena.ucsc.edu/, accessed on 1 September 2023). The comprehensive clinical characteristics of patients are outlined in Appendix A. RNA sequencing (RNA-seq) datasets were normalized using DESeq2 normalization, and data were stratified into low- and high-expression groups based on cutoff points established using the Evaluate Cutpoints software. The cutoff values for positive (high) and negative (low) mRNA expression of UBB, UBC, and CTNNB1 were as follows: <15.32; ≥15.32, <16.14; ≥16.14, <14.04; ≥14.04, respectively.

#### 2.4.2. The Gene Expression Omnibus

The microarray gene expression datasets GSE46602, GSE69223, GSE32571, GSE79021, GSE55945, and GSE14206, available from the Gene Expression Omnibus (GEO) repository, were downloaded through the ShinyGEO web-based application (http://gdancik.github.io/shinyGEO/, accessed on 13 October 2023) [19]. Box plots were used to depict the distribution of gene expression levels. The Mann–Whitney test or Wilcoxon test or unpaired *t*-test or paired *t*-test was used where appropriate to assess whether the expression difference was statistically significant (*p* < 0.05). In addition, the GSE46602 dataset, containing laser-microdissected benign prostate glands as well as laser-microdissected prostate tumor tissues and reporting recurrence-free survival (RFS) data for 36 patients, was retrieved, and the Kaplan–Meier curves were plotted using SPSS (version 28.0, IBM Corporation, Armonk, NY, USA). For survival analysis, the gene expression levels were dichotomized by the established cutoff point into high and low expression groups using the Evaluate Cutpoints software (UBB: 13.2; UBC: 13.03; β-Catenin: 11.23). If several probes were mapped to the UBB, UBC, or CTNNB1, the mean value was utilized as the final expression of these genes.

### 2.5. Statistical Analysis

All statistical analyses were conducted using GraphPad Prism v10.01 (GraphPad Software, La Jolla, CA, USA) and SPSS version 29.0 software (IBM Corporation, Armonk, NY, USA). Data normality was assessed using the Shapiro–Wilk test. The Mann–Whitney test was employed for comparing continuous variables, while the significance of clinical factors was evaluated using a two-tailed Chi-squared test or Fisher’s exact test. Survival curves were generated using the Kaplan–Meier method, and differences were assessed using a log-rank test. Univariable and multivariable Cox proportional hazards regression analyses were performed to estimate hazard ratios (HRs) and their corresponding 95% confidence intervals (CIs). Multivariable Cox proportional hazards models were used to test statistical independence of the predictors that showed the significance (*p* < 0.20) in the univariable analysis. These models were tested for proportional hazards assumptions based on Schoenfeld residuals and visual inspection of log (−log survival time) versus time plots; no model violated these assumptions. Statistically significant results were defined as those with *p*-values less than 0.05.

## 3. Results

### 3.1. Comparison of Protein Expression in Tumor Tissue and Normal Tissue and Its Association with ClinicalPathological Characteristics

In Figure 1, representative staining images for UBB, UBC, and β-Catenin expression in three distinct tissue types are presented: normal adjacent tissues, tumor tissues, and lymph node metastatic tissues. The results demonstrated a significantly higher UBB expression in tumor tissues compared to normal adjacent tissues (*p* = 0.001, Figure 2). Furthermore, we also observed a higher UBB expression in lymph node metastatic tissues compared to tumor tissues (*p* < 0.001, Figure 2) and a significant difference in UBB expression between metastatic tissues and normal tissues (*p* < 0.001, Figure 2). According to the cutoff points established by the Evaluate Cutpoints software, UBB expression was absent in 36 cases (53.7%), while 31 (46.3%) cases demonstrated the presence of UBB expression in tumor tissue. A significant association between UBB expression and age was observed (*p* = 0.0147). No other correlations between UBB expression and clinical-pathological data were identified (Table 1). No statistically significant correlations were observed between the analyzed proteins.

Similarly, the analysis of UBC expression revealed notable findings. We observed a significantly higher UBC expression in tumor tissues when compared to normal adjacent tissues (*p* = 0.0318, Figure 2). Furthermore, there was a notable increase in UBC expression in lymph node metastatic tissues in comparison to tumor tissues (*p* < 0.001, Figure 2). Significantly different UBC expression levels were also noted between metastatic tissues and normal tissues (*p* < 0.001, Figure 2). Based on the cutoff points determined, 41 (61.2%) cases demonstrated absence of expression, whereas 26 (38.8%) cases displayed presence of expression. A significant association was demonstrated between expression of UBC and pT status (*p* = 0.0098), and a marginally significant association was observed between UBC expression and Gleason score (*p* = 0.060). No further associations were noted between UBC expression and clinical-pathological data (Table 1). The results unveiled a significantly reduced expression of β-Catenin in tumor tissues when compared to normal adjacent tissues (*p* < 0.0001, Figure 2). No significant difference in β-Catenin expression was noted between tissues with lymph node metastases and tumor tissues (*p* = 0.6714, Figure 2). There was a substantial reduction in expression in tissues with metastases in contrast to normal tissues (*p* < 0.001, Figure 2). A normal pattern of β-Catenin (intense membranous staining) was observed in 37 (55.2%) cases, while loss of staining at the cell membrane was evident in 30 (44.8%) cases. No associations were observed between β-Catenin expression and clinical-pathological data (Table 1).

### 3.2. The Association between Expression of UBB, UBC, and β-Catenin Proteins and Patient Survival

Kaplan–Meier survival curve analysis revealed that patients with UBB expression in the cell nucleus had significantly lower (overall survival) OS rates compared to patients without nuclear expression (median OS: 2243 days vs. undefined days, *p* = 0.04, Figure 3A). The univariate Cox analysis demonstrated that the presence of UBB protein expression was predictive of an adverse OS (HR 2.22, 95% CI 1.00–4.91; *p* = 0.049; Table 2). In the multivariate Cox proportional hazards model, UBB protein expression emerged as an independent prognostic factor (HR 2.74, 95% CI 1.16–6.45; *p* = 0.022; Table 2).

Survival analysis demonstrated that PC patients with UBC expression had lower OS rates compared to patients without UBC expression in the nucleus (median OS: 1972 days vs. undefined days, *p* = 0.03, Figure 3B). The univariate Cox analysis revealed that the presence of UBC protein expression was associated with an unfavorable OS prognosis (HR 2.29, 95% CI 1.06–4.96; *p* = 0.035; Table 2). However, in the multivariate Cox proportional hazards model, UBC protein expression did not remain an independent prognostic factor (HR 1.39, 95% CI 0.61–3.17; *p* = 0.435; Table 2).

Kaplan–Meier survival analysis also revealed a significant association between abnormal β-Catenin expression and shorter survival in PC patients (median OS: 1936 days vs. undefined days, *p* = 0.02, Figure 3C). The univariate Cox analysis revealed that abnormal β-Catenin protein expression was associated with an unfavorable OS prognosis (HR 2.64, 95% CI 1.11–6.29; *p* = 0.028; Table 2). In the multivariate Cox proportional hazards model, β-Catenin protein expression remained a significant and independent prognostic factor (HR 2.60, 95% CI 1.05–6.44; *p* = 0.038; Table 2).

### 3.3. Overall Survival Analysis according to the Combined Biomarker Expression

After establishing the significance of the individual proteins as prognostic markers, we further investigated the impact of their combined expression on OS within our study cohort. The Kaplan–Meier analysis unveiled a significantly shorter OS in patients with concurrent expression of UBB and abnormal β-Catenin (median OS: 1342 days) compared to those with no UBB expression and normal β-Catenin (median OS: undefined days) as well as those with other combinations (median OS: 2243 days), e.g., simultaneous expression of UBB and normal β-Catenin or no UBB expression and abnormal β-Catenin (*p* = 0.008, Figure 4A). The Kaplan–Meier analysis similarly evaluated the expression of UBC and β-Catenin. It revealed a significantly shorter OS in patients with concurrent expression of UBC and abnormal β-Catenin (median OS: 1972 days) compared to those with no UBC expression and normal β-Catenin (median OS: undefined days) but not with other combinations (median OS: 1936 days), such as simultaneous expression of UBC and normal β-Catenin or no UBC expression and abnormal β-Catenin (*p* = 0.007, Figure 4B). Regrettably, the execution of a Cox regression analysis was precluded by the diminutive number of events within UBBabsent/β-Cateninnormal and UBCabsent/β-Cateninnormal subgroups of patients, which experienced remarkably improved survival compared to others.

In order to assess the interaction between UBB/β-Catenin, a dual-immunohistochemistry technique was utilized to visualize the mutual relationships between the markers of interest. This approach was employed to gain insights into the potential interplay and co-expression patterns of UBB/β-Catenin within the cellular context. Representative images illustrating the results of this dual-immunohistochemistry technique can be found in Figure 5.

### 3.4. Comparison of mRNA Expression in Tumor Tissue and Normal Tissue and Its Association with Clinical-Pathological Characteristics 

The results of the analyses conducted for the TCGA cohort revealed a significantly higher mRNA expression of UBB, UBC, and CTNNB1 in tumor tissues compared to normal tissues (*p* < 0.0001 for all, Figure 6). Based on the cutoff points determined by the Evaluate Cutpoints software, there were 195 cases (39.39%) with high UBB mRNA expression and 300 cases (60.61%) with low levels. Significant associations were observed between UBB mRNA expression and age (*p* = 0.0211) as well as between UBC mRNA expression and Gleason score (*p* = 0.025) and pN status (*p* = 0.0473). For UBC mRNA expression, 255 cases (51.52%) exhibited high levels, and 240 cases (48.48%) demonstrated low levels. As for CTNNB1 mRNA expression, 226 cases (45.66%) had high levels, while 269 cases (54.34%) had low levels. No correlations between CTNNB1 mRNA expression and clinical-pathological data were identified (Table 3). Statistical analysis of the Spearman correlation coefficient revealed a strong positive correlation between UBB mRNA and UBC mRNA (r = 0.5120; *p* < 0.0001). Similarly, the Spearman correlation coefficient showed a positive correlation between UBB mRNA and CTNNB1 mRNA (r = 0.1258; *p* = 0.005) and a positive correlation between UBC mRNA and CTNNB1 mRNA (r = 0.3005; *p* < 0.0001).

In contrast, the expression levels of UBB, UBC, and CTNNB1 in tumor versus normal tissues varied between the GEO cohorts, ranging from being unchanged to being slightly overexpressed or downregulated (Appendix A). 

### 3.5. The Association between the mRNA Expression of UBB, UBC, and CTNNB1 and Patient Survival

Survival analysis of the TCGA cohort indicated that the expression of UBB and CTNNB1 was not significantly correlated with OS (Figure 7A,C). Kaplan–Meier survival analysis demonstrated a significant association between low UBC expression and shorter survival in PC patients (median OS: undefined days vs. 3502 days, *p* = 0.003, Figure 7B). The Cox proportional hazard analysis was not carried out for the TCGA group due to an inadequate number of observed events. The limited number of specific outcomes in this group precluded the reliable application of this statistical method.

In contrast, survival analysis of PC patients from the GSE46602 series demonstrated that high expression levels of UBB and UBC were significantly associated with shorter RFS (*p* = 0.019 and *p* = 0.042, respectively; Appendix A). Furthermore, although there was a trend toward association of CTNNB1 low expression with worse RFS, this was not statistically significant (*p* = 0.119; Appendix A).

## 4. Discussion

Ubiquitination, a complex multistep process, hinges on the collaborative efforts of three distinct classes of enzymes, namely E1 (activating enzyme), E2 (coupling enzyme), and E3 (ligases). This orchestrated sequence begins with E1, the ubiquitin-activating enzyme, which initiates the process by binding to the C-terminal glycine residue of a ubiquitin monomer and an ATP molecule. This interaction leads to the formation of a crucial intermediate known as ubiquitin-adenylate. Subsequently, the E2 enzymes come into play, transferring the ubiquitin from E1 to the target substrate. The substrate then engages with the ubiquitin protein ligase (E3). In the initial stages, the first ubiquitin molecule is typically linked to the substrate via an isopeptide bond between the C-terminal glycine of ubiquitin and the amine group (E-NH2) of a lysine residue on the substrate. This process recurs in multiple cycles, leading to the formation of a polyubiquitin chain. These insights into the mechanics of ubiquitination have prompted investigations into the molecules involved in various types of cancer, underlining the broad relevance of this regulatory pathway [20,21,22]. In the context of PC, the expressions of UBC, UBB, and β-Catenin may have significant importance in unraveling the intricacies of its pathogenesis. Anomalies in ubiquitin expression and disruptions in ubiquitination can disturb the delicate balance of protein degradation in prostate cancer, leading to the accumulation of proteins regulating cell growth. This accumulation, in turn, may contribute to the uncontrolled proliferation of cancer cells, which is a characteristic feature of tumor development. UBC represents one of the many forms of ubiquitin. Collaborating with other ubiquitin molecules, UBC forms ubiquitin chains crucial for tagging target proteins for degradation. Its role in prostate cancer is pivotal, as it directly influences the tagging of proteins for degradation [11,14,15]. Dysregulation of UBC expression can disturb the ubiquitin–proteasome system, affecting the timely removal of regulatory proteins. UBB, another player in this complex network, serves as one of the building blocks for ubiquitin. Together with its counterparts, it forms ubiquitin chains that contribute to the tagging process [11,12]. Disruptions in UBB expression can further exacerbate the dysregulation of ubiquitin-mediated protein degradation, intensifying the potential impact on prostate cell regulation and cancer progression. β-Catenin, a key component of the Wnt signaling pathway, is tightly regulated under normal cellular conditions, influencing processes such as cell proliferation and differentiation [9,10,13,14,15]. However, anomalies in the Wnt pathway and β-Catenin instability in prostate cancer can lead to uncontrolled cell growth. The accumulation of β-Catenin may activate genes associated with cell proliferation, significantly accelerating the initiation and progression of prostate cancer. In the context of this study, our principal aim was to investigate the intricate interplay among the expression profiles of UBB, UBC, and β-Catenin and their associations with diverse clinicopathological variables. We placed a special emphasis on understanding how these expression patterns relate to the survival outcomes of patients with PC. Our research sought to determine the potential of these factors as independent prognostic markers in PC. Additionally, we adopted a complementary approach to assess their combined prognostic capability.

In the course of our investigation, we consistently observed a significant upregulation in the expression of both UBB and UBC within PC tissue in comparison to adjacent normal tissue. Notably, this upregulation was even more pronounced in lymph node metastases, underscoring the critical role of increased UBB and UBC expression in the context of PC progression. This suggests that these elevated expression levels may serve as indicators of an underlying mechanism associated with the aggressive nature of PC, particularly in cases involving lymph node metastases. Additionally, these findings hint at a potential association between this observed upregulation and an increased prevalence of metabolic alterations within cancerous tissues as opposed to their healthy counterparts. It is worth noting that these observations align with the existing literature that has reported the overexpression of UBB and UBC in various cancer types, including PC and non-small-cell lung cancer [23,24]. Moreover, our findings are consistent with previous research. For instance, UBB silencing has been shown to significantly decrease the proliferation rate of neuroblastoma, hepatocarcinoma, breast, and PC cells [25]. These results are in line with the study by Tang et al., who found that knockdown of UBC and UBB inhibited cell growth and weakened radioresistance in lung cancer cells, both in vitro and in vivo [23]. This collective body of evidence highlights the potential therapeutic significance of targeting UBB and UBC in cancer treatment and emphasizes the multifaceted factors contributing to the aggressive behavior of various cancer types.

The observed associations between the levels of UBB and UBC proteins and the OS of PC patients are indeed intriguing. These findings provide valuable insights into the potential prognostic significance of UBB and UBC in the context of PC. Understanding these correlations holds clinical relevance and has the potential to impact the diagnosis and management of PC. What is particularly noteworthy is that patients who exhibited high expression levels of both UBB and UBC had notably poorer OS compared to individuals with lower expression levels. This suggests that the assessment of UBB and UBC may serve as a comprehensive and informative prognostic factors, enabling the identification of high-risk patient groups that might require more aggressive treatment strategies or closer monitoring. Furthermore, the identification of UBB as an independent prognostic marker is a significant discovery. This implies that UBB can provide valuable information regarding the prognosis of PC patients, independently of other clinical variables. Independent markers of this nature are particularly valuable as they can assist clinicians in making more precise prognostic assessments and treatment decisions.

The results of our study shed light on the compelling potential of UBB and UBC as prognostic markers in the context of PC. These findings hold significant clinical relevance, as they have the capacity to influence the diagnosis and management of this complex disease. In essence, the findings of our study underscore the potential of UBB and UBC as prognostic markers for PC and emphasize the importance of considering them in the overall clinical evaluation of PC patients. In addition to their role as prognostic markers, our findings suggest that UBB and UBC may also serve as potential tools for diagnosing metastatic spread in PC. This potential dual utility enhances the clinical significance of these markers, as the early detection of metastases is crucial for selecting appropriate treatment strategies and improving patient outcomes.

Our results demonstrated a statistically significant reduction in membranous β-Catenin expression within tumor tissues when compared to their normal adjacent counterparts. This finding underscores the potential relevance of β-Catenin in the context of PC development, suggesting a shift in β-Catenin expression within the disease’s progression. Importantly, these observations are in line with reports from other researchers [26,27]. It is worth noting that β-Catenin plays a pivotal role in the Wnt signaling pathway, which is associated with the process of Epithelial–Mesenchymal Transition (EMT) [28]. EMT is a biological process in which epithelial cells transform into mesenchymal cells, a phenomenon typically seen in developmental processes but also cancer. EMT is linked to the loss of cell adhesion, increased cell mobility, invasion capability, and the formation of cancer metastases [29]. Therefore, the reduction in membranous β-Catenin expression in PC may indicate a potential loss of cell adhesion and indicate Wnt pathway activation and the EMT process. This, in turn, may contribute to a more aggressive nature of the tumor, increased capacity for spreading and invading surrounding tissues, and the formation of metastases [30,31]. However, no significant difference in β-Catenin expression was observed between lymph node metastatic and tumor tissues. These findings suggest that the reduced expression of β-Catenin is a characteristic of the tumor tissues themselves, but the presence of lymph node metastases does not further affect the expression of this protein. Based on our analysis, abnormal β-Catenin expression is robustly linked to shorter survival in PC patients and represents a significant, independent prognostic factor for patient outcomes. Multiple studies have consistently demonstrated that reduced or unstable expression of membranous β-Catenin is correlated with poorer patient survival [26,32,33,34]. This implies that the loss of membranous β-Catenin may contribute to a more aggressive disease course. The correlation between β-Catenin expression and patient outcomes highlights the potential clinical significance of β-Catenin as a prognostic marker in various types of cancer.

A potential mechanistic linkage between the expression level of β-Catenin and the metabolic pathways implicated in cancer progression is suggested. The observation that high concentrations of glucose specifically increase the levels of c-Myc and β-Catenin in hepatocellular cells, as described by Chouhan et al., underscores the potential link between glucose metabolism and the activation of signaling pathways associated with proliferation [35]. Activation of the canonical Wnt signaling pathway by high glucose levels, leading to the transcription of β-Catenin-responsive genes in HCC cells, as reported, emphasizes the significance of elucidating molecular interactions between glucose metabolism and signaling pathways in cancer progression [35]. Findings from studies by Chouhan et al. and Nguyen et al. highlight the importance of understanding the relationship between the metabolism of cancer cells and the activity of signaling pathways in the context of PC development and treatment [36,37]. Both articles indicate the therapeutic potential arising from the regulation of signaling pathways and the metabolism of cancer cells. The discovery of potential therapeutic targets, such as TNK2/ACK1 or the signaling nexus pY-SREBF1/H2A-K130ac, suggests the possibility of developing new treatment strategies for PC patients, especially those affected by resistance to traditional therapies [36,37].

Finally, taking into consideration the functional relationship between UBB, UBC, and β-Catenin, we initiated an inquiry to evaluate the cumulative impact of the co-expression of these proteins on the OS of patients with PC. The Kaplan–Meier survival analysis revealed that patients with tumors characterized by the simultaneous expression of UBB and aberrant β-Catenin exhibited the poorest OS outcomes. Notably, the concurrent expression of these two markers displayed a superior predictive capacity for patient survival when compared to assessing each marker in isolation. Regrettably, our attempts to conduct a Cox regression analysis were constrained by the limited group sizes present in our dataset. Our analytical approach encompassed a multitude of prognostic determinants, and the inherent instability or scarcity of data points in certain intersections of these determinants precluded the undertaking of a Cox regression analysis. It is crucial to underscore that Cox regression, being a method of heightened statistical complexity, necessitates a more substantial dataset to yield results of requisite statistical robustness. Nevertheless, it is paramount to recognize that the insights gleaned from the Kaplan–Meier analyses have furnished invaluable perspectives into the intricate interplay of these determinants on the overarching construct of OS within the scope of our investigation. Our analyses have illuminated the potential value of employing immunohistochemistry to appraise the combined expression of UBB and β-Catenin as a more efficacious approach in prognosticating the outcomes of PC patients, surpassing the utility of individual markers. The comprehensive panel we have meticulously crafted serves as a valuable resource for clinicians, equipping them with the knowledge needed to make well-informed decisions concerning the prognosis of PC. With the ability to predict patient outcomes with greater accuracy, this panel can significantly enhance the clinical management of PC.

The outcomes derived from our cohort bring to light a noteworthy disparity between mRNA and protein, emphasizing that they do not consistently align in terms of expression patterns and/or prognostic values. Observed discrepancies were mostly in expression statusdetected between normal and tumor tissues using immunohistochemistry versus gene microarray, whereby, based on the latter, we failed to find reproducible expression patterns among the analyzed GEO datasets, as the expression levels of UBB, UBC, and CTNNB1 rangedfrom being unchanged to being slightly overexpressed or downregulated. These differences are possibly due in part to limitations of small sample sizes, the origin of normal tissue for comparative analyses, different microarray platforms and analysis techniques, varying sampling methods, difficulties related to probe and probe set identity as well as actual biological differences. It is also essential to underline that the expression of UBB and UBC extends beyond the confines of the cell’s nucleus, encompassing the cytoplasm as well. Similarly, it is vital to emphasize that β-Catenin is present in the cytoplasm, cell membrane, and nucleus. However, it is important to clarify that our primary focus in this study was on the nuclear component when assessing UBB and UBC, whereas for β-Catenin, our analysis was solely directed at the cell membrane. Furthermore, measurements of mRNA levels by bulk RNA-seq or microarray are summed over populations of different cell types in a tissue, thus they are a reflection of the average gene expression in the entire FFPE slice. In contrast, by employing immunohistochemistry, one can meticulously detect and localize a specific protein to specific cell types. This approach enables us to gain valuable insights into the intricate interplay of these proteins, a level of detail that mRNA analysis alone cannot provide. In this context, the interesting finding of our study is that the mRNA expression status by microarray-based quantitation of laser capture microdissected cancer cell populations yielded similar relationships with patient survival as the immunohistochemical assessment of protein abundance; however, IHC still related better to prognosis. Our study highlights the pivotal role of immunohistochemistry as a reliable tool for the development of an intricate panel, offering precise prognostic insights for patients with PC. Immunohistochemistry, an economical method for visualizing specific proteins within tissue samples, emerges as a cost-effective and efficient solution.

It is crucial to acknowledge that our study has a significant limitation related to the size of the cohort we examined. A small cohort inherently means that the number of patients included in the study is limited, which can affect the ability to draw statistically robust conclusions. This limitation underscores the need for caution when interpreting the results and highlights the importance of conducting further research with larger cohorts to validate and strengthen our findings.

## 5. Conclusions

In summary, our study underscores the multifaceted significance of UBB, UBC, and β-Catenin as promising prognostic markers and potentially valuable diagnostic tools for PC. Elevated levels of UBB and UBC, particularly in lymph node metastases, suggest a higher risk of disease progression, potentially guiding more aggressive treatment approaches. Reduced membranous β-Catenin expression may require tailored interventions to address the disease’s aggressiveness. Integrating these markers into routine assessments could enhance prognostic accuracy, allowing for personalized treatment strategies. Additionally, our study highlights the importance of immunohistochemistry for assessing protein expression, providing valuable prognostic insights. Overall, leveraging these markers could improve outcomes and quality of care for PC patients.

## Figures and Tables

**Figure 1 cancers-16-00902-f001:**
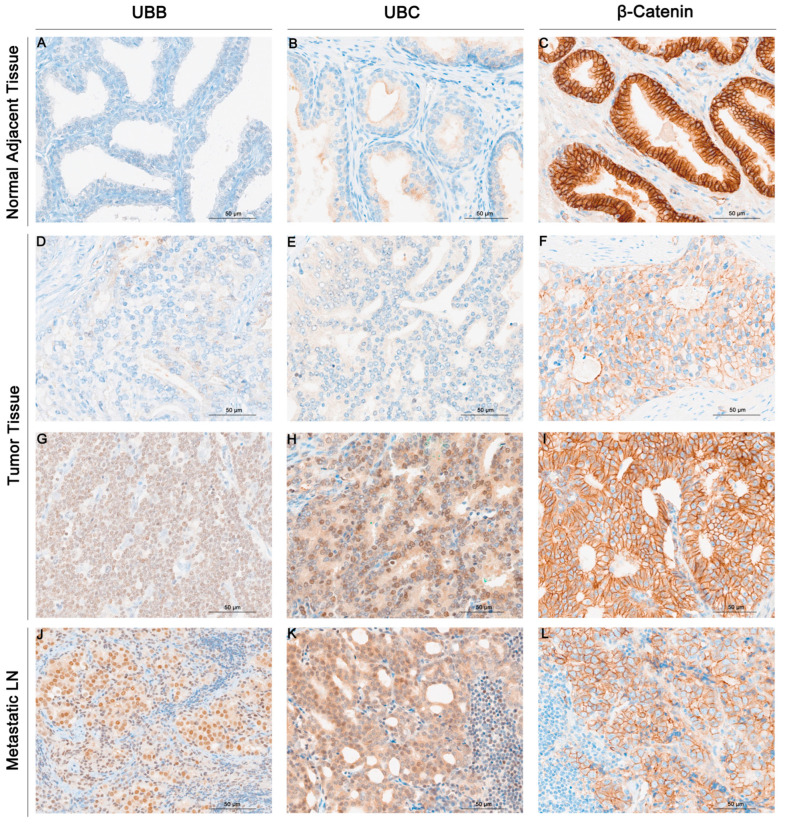
Exemplar images illustrating the immunohistochemical staining of UBB (**A**,**D**,**G**,**J**), UBC (**B**,**E**,**H**,**K**), and β-Catenin (**C**,**F**,**I**,**L**) in adjacent normal tissues, prostate cancer, and metastatic lymph nodes. UBB staining in adjacent normal tissues (**A**); absence of UBB staining (**D**); presence of UBB staining in prostate cancer (**G**); UBB staining in metastatic lymph nodes (**J**); UBC staining in adjacent normal tissues (**B**); absence of UBC staining (**E**); presence of UBC staining in prostate cancer (**H**); UBC staining in metastatic lymph nodes (**K**); β-Catenin staining in adjacent normal tissues (**C**); weak staining (**F**) and strong staining (**I**) for β-Catenin in prostate cancer; β-Catenin staining in metastatic lymph nodes (**L**). The original magnification is 20×.

**Figure 2 cancers-16-00902-f002:**
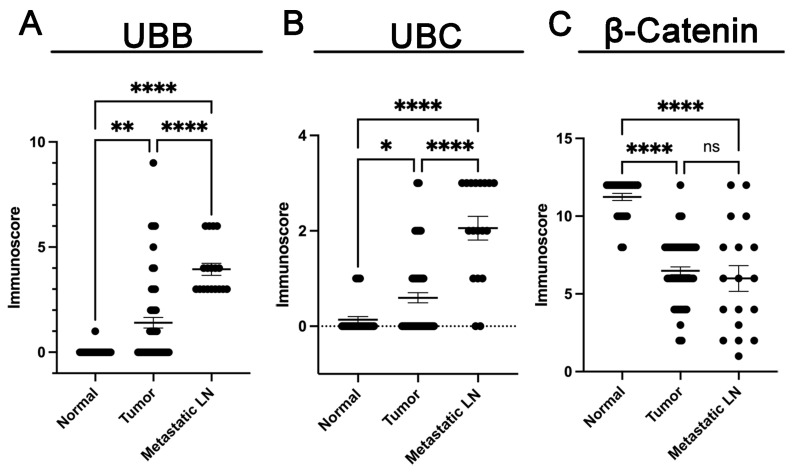
Comparison of immunohistochemical expression of UBB (**A**), UBC (**B**), and β-Catenin (**C**) in normal prostate tissues (adjacent to the tumor), prostate cancer tissues, and prostate cancer metastases to lymph nodes. Errors are represented as Standard Error of the Mean (SEM). The symbol * indicates a highly significant difference, with *p* < 0.05; ** indicates a highly significant difference, with *p* < 0.01; **** signifies an extremely high level of significance, with *p* < 0.0001; “ns” (not significant) suggests that there is no statistically significant difference between the compared groups, with *p* > 0.05.

**Figure 3 cancers-16-00902-f003:**
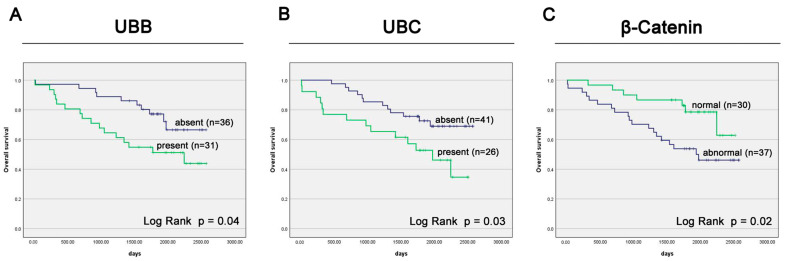
Kaplan–Meier survival curves stratified by expression of UBB (**A**), UBC (**B**), and β-Catenin (**C**) in prostate adenocarcinoma. Protein expression was assessed through immunohistochemistry, and the calculated Immunoreactive Score (IRS) results were categorized into two groups, as specified in the Materials and Methods section.

**Figure 4 cancers-16-00902-f004:**
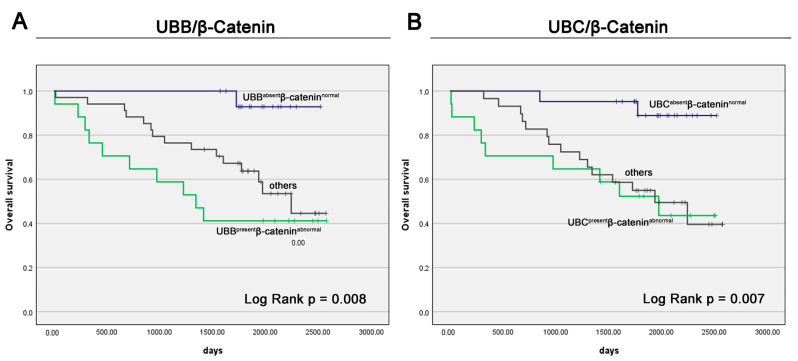
Kaplan–Meier survival curves stratified by the expression of the combination of the panel UBB/β-Catenin (**A**) and UBC/β-Catenin (**B**). Protein expression was assessed through immunohistochemistry, and the calculated Immunoreactive Score (IRS) results were categorized into two groups, as specified in the Materials and Methods section.

**Figure 5 cancers-16-00902-f005:**
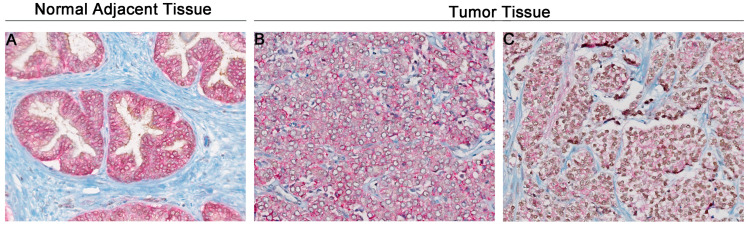
Images illustrating co-staining of UBB/β-Catenin (UBB: brown nuclei; β-Catenin: pink membranes) in adjacent normal tissues and prostate cancer. UBB/β-Catenin staining in adjacent normal tissues (**A**); UBB/β-Catenin staining in prostate cancer: absence of UBB staining and strong staining for β-Catenin (**B**); UBB/β-Catenin staining in prostate cancer: presence of UBB staining and weak staining for β-Catenin (**C**). The original magnification is 20×.

**Figure 6 cancers-16-00902-f006:**
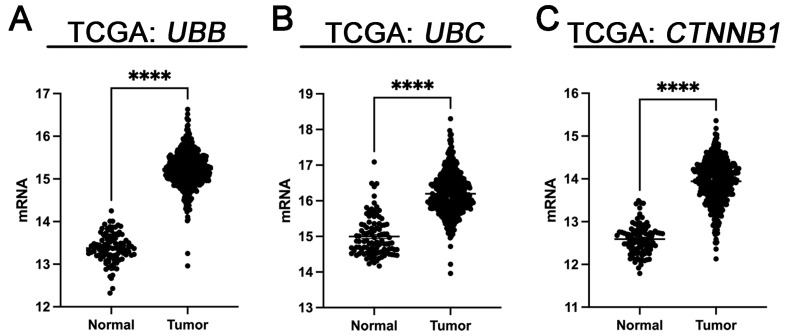
mRNA expression of UBB (**A**), UBC (**B**), and CTNNB1 (β-Catenin) (**C**) in tumor and normal tissues in PC. Errors are represented as Standard Error of the Mean (SEM). The symbol **** signifies statistical significance, with *p* < 0.0001.

**Figure 7 cancers-16-00902-f007:**
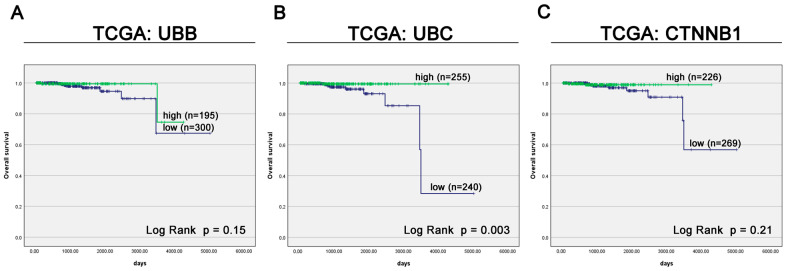
Kaplan–Meier survival curves stratified by UBB (**A**), UBC (**B**), CTNNB1 (β-Catenin) (**C**), mRNA levels in TCGA cohort.

**Table 1 cancers-16-00902-t001:** Associations between UBB, UBC, and β-Catenin and clinicopathological characteristics in our cohort.

Variables	*n* = 67	UBB		*p* Value	UBC		*p* Value	β-Catenin		*p* Value
		Absent	Presence		Absent	Presence		Abnormal	Normal	
		*n* = 36	*n* = 31		*n* = 41	*n* = 26		*n* = 37	*n* = 30	
Age (years)										
≤65	32 (47.76)	12 (37.50)	20 (62.50)	0.0147	20 (62.50)	12 (37.50)	>0.9999	17 (53.13)	15 (46.88)	0.8084
>65	35 (52.24)	24 (68.57)	11 (31.43)		21 (60.00)	14 (40.00)		20 (57.14)	15 (42.86)	
Gleason score										
GS 6	3 (4.48)	1 (33.33)	2 (66.67)	0.6917	3 (100.00)	0 (0)	0.06	3 (100.00)	0 (0)	0.5407
GS 7	35 (52.24)	21 (60.00)	14 (40.00)		24 (68.57)	11 (31.43)		18 (51.43)	17 (48.57)	
GS 8	11 (16.42)	5 (45.45)	6 (54.55)		5 (45.45)	6 (54.55)		7 (63.64)	4 (36.36)	
GS 9	18 (26.87)	9 (50.00)	9 (50.00)		9 (50.00)	9 (50.00)		9 (50.00)	9 (50.00)	
Grade group										
group 1	3 (4.48)	1 (33.33)	2 (66.67)	0.7996	3 (100.00)	0 (0)	0.2908	3 (100.00)	0 (0)	0.5916
group 2	11 (16.42)	6 (54.55)	5 (45.45)		7 (63.64)	4 (36.36)		5 (45.45)	6 (54.55)	
group 3	24 (35.82)	15 (62.50)	9 (37.50)		17 (70.83)	7 (29.17)		13 (54.17)	11 (45.83)	
group 4	11 (16.42)	5 (45.45)	6 (54.55)		5 (45.45)	6 (54.55)		7 (63.64)	4 (36.36)	
group 5	18 (26.87)	9 (50.00)	9 (50.00)		9 (50.00)	9 (50.00)		9 (50.00)	9 (50.00)	
pT status										
T2	9 (13.43)	5 (55.56)	4 (44.44)	>0.9999	9 (100.00)	0 (0)	0.0098	5 (55.56)	4 (44.44)	>0.9999
T3-T4	58 (86.57)	31 (53.45)	27 (46.55)		32 (55.17)	26 (44.83)		32 (55.17)	26 (44.83)	
pN status										
Nx	1									
N0	43 (65.15)	24 (55.81)	19 (44.18)	0.6094	30 (69.77)	13 (30.23)	0.1111	27 (62.79)	16 (37.21)	0.0761
N1	23 (34.85)	11 (47.83)	12 (52.17)		11 (47.83)	12 (52.17)		9 (39.13)	14 (60.87)	
PSA										
x	2									
≤10 ng/mL	27 (41.54)	14 (51.85)	13 (48.15)	>0.9999	17 (62.69)	10 (37.04)	0.8019	15 (55.56)	12 (44.44)	>0.9999
>10 ng/mL	38 (58.46)	21 (55.26)	17 (44.74)		22 (57.89)	16 (42.11)		20 (52.63)	18 (47.37)	

**Table 2 cancers-16-00902-t002:** Univariate and multivariate analysis of prognostic factors using the Cox proportional hazard model in our own cohort.

Variable	n/EPV	Univariable Analysis				Multivariable Analysis			
		HR	95% CI		*p* Value	HR	95% CI		*p* Value
			L	U			L	U	
UBB (present vs. absent)	31/16_36/10	2.22	1.00	4.91	0.049	2.74	1.16	6.45	0.022
UBC (present vs. absent)	26/14_41/12	2.29	1.06	4.96	0.035	1.39	0.61	3.17	0.435
β-Catenin (abnormal vs. normal)	37/19_30/7	2.64	1.11	6.29	0.028	2.60	1.05	6.44	0.038
Age (≥69 vs. <69)	25/12_42/14	1.70	0.78	3.68	0.179	1.83	0.79	4.23	0.161
Gleason group (5 vs. 1–4)	18/10_49/16	1.82	0.82	4.01	0.140	2.14	0.94	4.89	0.071
pT (T3–T4 vs. T2)	58/23_9/3	1.25	0.37	4.19	0.719	-	-	-	-
pN (N1 vs. N0)	23/10_44/16	1.35	0.61	2.98	0.464	-	-	-	-
TNM stage (IV vs. II–III)	23/10_44/16	1.35	0.61	2.98	0.464	-	-	-	-
PSA (≥13.4 vs. <13.4)	31/15_36/11	1.81	0.83	3.95	0.138	1.64	0.73	3.70	0.231

**Table 3 cancers-16-00902-t003:** Associations between UBB, UBC, and CTNNB1 and clinicopathological characteristics in TCGA cohort.

Variables	*n* = 495	UBB		*p* Value	UBC		*p* Value	CTNNB1		*p* Value
		Low	High		Low	High		Low	High	
		*n* = 300 (%)	*n* = 195 (%)		*n* = 240	*n* = 255		*n* = 269	*n* = 226	
Age (years)										
≤60	222 (44.85)	122 (54.95)	100 (45.05)	0.0211	103 (46.40)	119 (53.60)	0.4168	117 (52.70)	105 (47.30)	0.5263
>60	273 (55.15)	178 (65.20)	95 (34.80)		137 (50.18)	136 (49.82)		152 (55.68)	121 (44.32)	
Gleason score										
GS 6	45 (9.09)	26 (57.78)	19 (42.22)	0.632	22 (48.89)	23 (51.11)	0.025	29 (64.44)	16 (35.56)	0.4587
GS 7	247 (49.90)	150 (60.73)	97 (39.27)		129 (52.23)	118 (47.77)		130 (52.63)	117 (47.37)	
GS 8	64 (12.93)	44 (68.75)	20 (31.25)		36 (56.25)	28 (43.75)		36 (56.25)	28 (43.75)	
GS 9	135 (27.27)	78 (57.78)	57 (42.22)		50 (37.04)	85 (62.96)		73 (54.07)	62 (45.93)	
GS 10	4 (0.81)	2 (50.00)	2 (50.00)		3 (75.00)	1 (25.00)		1 (25.00)	3 (75.00)	
pT status										
x	7									
T2	187 (38.32)	119 (63.64)	68 (36.36)	0.2962	97 (51.87)	90 (48.13)	0.2645	108 (57.75)	79 (42.25)	0.3996
T3–T4	301 (61.68)	177 (58.80)	124 (41.20)		140 (46.51)	161 (53.49)		161 (53.49)	140 (46.51)	
pN status										
x	73									
N0	342 (81.04)	210 (61.40)	132 (38.60)	>0.9999	171 (50.00)	171 (50.00)	0.0473	181 (52.92)	161 (47.08)	>0.9999
N1	80 (18.96)	49 (61.25)	31 (38.75)		30 (37.50)	50 (62.50)		42 (52.50)	38 (47.50)	
PSA										
x	56									
<4 ng/mL	368 (83.83)	222 (60.33)	146 (39.67)	0.1841	172 (46.74)	196 (53.26)	0.2431	198 (53.80)	170 (46.20)	0.5176
>4 ng/mL	71 (16.17)	49 (69.01)	22 (30.99)		39 (54.93)	32 (45.07)		35 (49.30)	36 (50.70)	

## Data Availability

The datasets generated and analyzed during the current study can be obtained from the corresponding author upon reasonable request.

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
