# Peer review of "Ubiquitin B, Ubiquitin C, and β-Catenin as Promising Diagnostic and Prognostic Tools in Prostate Cancer"

_cancers, 2024, doi:10.3390/cancers16050902_

Round 1
Reviewer 1 Report
Comments and Suggestions for Authors
The manuscript titled " UBB, UBC, and β-Catenin as Promising Diagnostic and Prognostic Tools in Prostate Cancer" provides a comprehensive summary of the study on ubiquitination processes and expression regulation in prostate cancer.
However, there are following major concerns with the manuscript which need to be addressed:
1. The authors need to provide scales bares for all the photographic figures.
2. Could you elaborate on the specific methodologies used for the analysis of expression profiles and their correlation with clinical and histopathological data?
3. Is the regulation of Beta-catenin often associated with Wnt3a? Have the authors examined the levels of Wnt3a in human prostate cancer samples using IHC or mRNA levels?
4. The ubiquitination of beta-catenin is frequently linked to a decrease in its cytosolic and nuclear functionality. Could the authors elaborate on the reasons behind the observed increase in UBB and UBC levels in tumors and metastatic lymph node samples? Additionally, it would be valuable for the authors to furnish the nuclear staining score of beta-catenin in IHC samples to elucidate this potential correlation.
5. The authors should provide further details on the implications of the study's conclusions, particularly in relation to specific patient demographics or other relevant considerations.
Reviewer 2 Report
Comments and Suggestions for Authors
This study relies on a limited patient cohort (67 individuals), potentially introducing interference in result interpretation. The assessment of immunohistochemical staining is not completely clear, particularly regarding the predominant scoring of Beta-catenin as either nuclear or cytoplasmic staining pattern in routine pathologist practice. Consequently, the described scoring system needs more clarity, particularly concerning the significance of a cut-point of 8, which requires a more detailed explanation.
To enhance understanding, figures depicting the immunohistochemical scoring system for β-Catenin, UBB, and UBC would be beneficial. As these three markers form the basis of the study's results, a thorough description of the scoring system, accompanied by illustrative figures and photos, is warranted.
A more explicit description of the group with lymph node metastases from primary tumors in the research is unclear; thus, clarification in the methodology section is essential. Are these lymph node metastases from the same patients as the primary tumor? Lymph nodes are resected during prostate resections or after some period when the tumor was recurrent?
Furthermore, the extent of discrepancies in Beta-catenin, UBB, and UBC expression between primary and metastatic tumors might enrich the study. Specifically, the number of patients exhibiting such disparities should be explicitly addressed.
Reviewer 3 Report
Comments and Suggestions for Authors
1. Authors need to provide the specific reasons for the significant expression of UBC, UBB and β-Catenin in PC.
2. Authors need to discuss the importance of UBC, UBB and β-Catenin in PC progression clearly. Are they involve any tumor suppression or progression pathways?
3. Authors need to provide scale bar in IHC figures.
Round 2
Reviewer 1 Report
Comments and Suggestions for Authors
In the revised manuscript titled " UBB, UBC, and β-Catenin as Promising Diagnostic and Prognostic Tools in Prostate Cancer" the authors have made commendable efforts to address most of the previous concerns, resulting in a convincing and valuable contribution to the understanding of the role of UBB, UBC, and β-Catenin in Prostate Cancer. While the manuscript has been substantially improved, I note that there is room for enhancing the discussion section, and the inclusion of pertinent studies such as PMID: 27272409, 37296155 and 35895804 would significantly enhance its overall impact. These studies, in particular, explore significant modifications in key apoptotic and Wnt signaling pathways, shedding light on alterations in proliferation pathways across various cancer types, including prostate cancer. By incorporating findings from these studies, the authors can substantially strengthen the foundation for their compelling discovery of unique and druggable targets in cancer. I recommend the publication of this article with minor revisions, primarily focusing on the addition of relevant literature and further enriching the discussion to provide a more comprehensive context for the study's significance.
Author Response
Dear Reviewer,
Thank you for your thorough evaluation of our revised manuscript titled "UBB, UBC, and β-Catenin as Promising Diagnostic and Prognostic Tools in Prostate Cancer." We appreciate your acknowledgment of the efforts we have put into addressing previous concerns and are delighted that you find our contribution valuable Thank you for your thorough evaluation of our revised manuscript titled "UBB, UBC, and β Promising Diagnostic and Prognostic Tools in Prostate Cancer." We appreciate your acknowledgment efforts we have put into addressing previous concerns and are delighted that you find our contribution valuable to the understanding of the role of UBB, UBC, and β Thank you for your thorough evaluation of our revised manuscript titled "UBB, UBC, and β Promising Diagnostic and Prognostic Tools in Prostate Cancer." We appreciate your acknowledgment efforts we have put into addressing previous concerns and are delighted that you find our contribution valuable to the understanding of the role of UBB, UBC, and β-Catenin in prostate cancer. We are grateful for your constructive feedback regardin g the discussion section. We agree that enhancing this We are grateful for your constructive feedback regarding the discussion section. We agree that enhancing this section with pertinent studies would further strengthen the impact of our findings. We have reviewed the studies you mentioned (PMID: 27272409, 37296155, and 35895804) and recognize their significance elucidating modifications in apoptotic and Wnt signaling pathways across various cancer types, including section with pertinent studies would further strengthen the impact of our findings. We have reviewed the studies you mentioned (PMID: 27272409, 37296155, and 35895804) and recognize their significance in elucidating modifications in apoptotic and Wnt signaling pathways across various cancer types, including section with pertinent studies would further strengthen the impact of our findings. We have reviewed the studies you mentioned (PMID: 27272409, 37296155, and 35895804) and recognize their significance elucidating modifications in apoptotic and Wnt signaling pathways across various cancer types, including prostate cancer. Thank you for considering our work for publication. Thank you for considering our work for publication.
